Potential applications of personality assessments to the management of non-human primates: a review of 10 years of study

Norman Max 1 2 enorman19@rvc.ac.uk
Rowden Lewis J. 3
http://orcid.org/0000-0002-7269-4625 Cowlishaw Guy 2
1 Royal Veterinary College , London , United Kingdom
2 Institute of Zoology, Zoological Society of London , London , United Kingdom
3 Zoological Society of London , London , United Kingdom
Barrett Louise
Electronic publication date: 2021 Sep 7
Publication date: 2021
Volume: 9
Electronic Location ID: e12044
Received 2021 Jan 27; Accepted 2021 Aug 2
Copyright: © 2021 Norman et al.
Copyright year: 2021
Copyright holder: Norman et al.
License: This is an open access article distributed under the terms of the Creative Commons Attribution License, which permits unrestricted use, distribution, reproduction and adaptation in any medium and for any purpose provided that it is properly attributed. For attribution, the original author(s), title, publication source (PeerJ) and either DOI or URL of the article must be cited.
License URL: https://creativecommons.org/licenses/by/4.0/

Keywords: Primate, Personality, Temperament, Animal welfare, Animal management

Funding: Royal Veterinary College Research England This research was funded by the Royal Veterinary College and by Research England. The funders had no role in study design, data collection and analysis, decision to publish, or preparation of the manuscript.

==============================
Studies of primate personality have become increasingly common over the past three decades. Recently, studies have begun to focus on the health, welfare and conservation implications of personality, and the potential applications of incorporating quantitative personality assessments into animal management programmes. However, this literature is dispersed across a multitude of settings and scientific disciplines. We conducted a review of nonhuman primate personality studies relevant to these issues published since 2010, following on from an earlier review. The databases ScienceDirect, PubMed and Web of Science were used to identify relevant articles. After eliminating irrelevant or duplicate papers, 69 studies were selected. Our review reveals that, while primate personality research is carried out on a range of species, there is strong taxonomic bias. While 28 species appeared within the reviewed literature, 52% of studies were carried out on just five species. Further, the most common research focus (43%) was validating new assessment methods or describing personality in different species, rather than exploring the links between personality and animal welfare using existing validated methods. However, among the remaining studies that did explore the role of animal personality in husbandry, health, and welfare, we identified progression towards integrating personality data into various aspects of animal management. Evidence suggests the assessment of personality may benefit social group management, enrichment practices, training protocols, health and welfare monitoring, and conservation planning for endangered species. We argue that further research which develops our understanding of primate personality and its influence in these areas will provide a valuable tool to inform animal management practices.

Introduction

Institutions which keep captive primates, whether as part of a zoological collection, breeding facility or research laboratory, strive to continually improve welfare and conditions for the animals in their care. One area of growing interest is incorporating individual behavioural differences into management plans. Evidence suggests that incorporating such differences could significantly improve management outcomes; for example, personality is associated with breeding success and pair compatibility in black rhinoceros (Carlstead, Mellen & Kleiman, 1999). While species differences in behavioural responses to captivity and management practices are recognised (Mason, 2010) the predominant focus in such approaches has been on behavioural differences related to age, sex, and rank. However, with the advent of the 21st century, the study of intraspecific behavioural differences has become an increasingly explored area of applied behavioural science.

“Temperament” or “Personality” is broadly described as individual differences in behavioural tendency that are consistent across time and contexts (Watters & Powell, 2012). Each “personality trait”, or “dimension”, is defined by Eysenck (1997) as a spectrum along which consistent individual differences in specific groups of behaviours can be quantitatively measured (Itoh, 2002). Primate personality dimensions are well described; personality assessments are available for several species including chimpanzees, gorillas, and rhesus macaques (Freeman & Gosling, 2010). Personality assessments in primates have been demonstrated to have construct validity (Freeman & Gosling, 2010), i.e. they measure what they purport to measure (Cronbach & Meehl, 1955; Campbell & Fiske, 1959; Itoh, 2002). Analyses of personality assessments have also been established as reliable (Freeman & Gosling, 2010); that is, consistent scores are produced when a subject is tested against the same assessment multiple times or by different assessors (Itoh, 2002).

The literature suggests that personality plays a role in influencing behaviour, social compatibility, health, and reproductive success of an array of taxa, not only primates (Tetley & O’Hara, 2012), and so may provide information to inform management practices and, potentially, increase their effectiveness. While individual character traits are often recognised by those who work closely with animals and considered in management decisions (Watters & Powell, 2012), investigating the potential of quantitative measurements of personality is still a new area of research. Potential applications include and are not limited to tailoring enrichment programmes to reduce stress and stereotypic behaviour more effectively (Gartner & Powell, 2012; Franks et al., 2013; Hopper et al., 2014), identifying individuals at higher risk of developing stress-related morbidities (Jin et al., 2013; Gottlieb et al., 2018), and improving the success of social introductions (Martin-Wintle et al., 2017).

While the possibilities are promising, practical application of individual personality data requires an understanding of the mechanisms by which personality predicts health and welfare outcomes. Despite primates being popular subjects of personality research (Freeman & Gosling, 2010), the field has, historically, been fragmented (Tetley & O’Hara, 2012). Published studies of primate personality encompass a broad spectrum of disciplines, including psychology, evolution, and zoology, which, while providing multiple perspectives, can make it challenging to synthesise their results. Bringing together this research is essential to prevent lines of investigation from becoming isolated (Tetley & O’Hara, 2012) and to generalise findings for real-world application, i.e. to draw broad conclusions which extend to species or settings outside of those within individual studies (Polit & Beck, 2010).

A previous review by Freeman & Gosling (2010) explored how the field of primate personality research has developed since the early 1900s. While this review identified an increasing interest in, and acceptance of, the scientific study of personality in nonhuman primates, the authors highlighted gaps in the literature. Freeman and Gosling recommended that future study should aim to examine and realise the potential practical uses of personality to the management of primates. The last paper to synthesize the results of the primate personality literature in this way was by Coleman (2012), who identified that, while the use of personality assessments held promise in areas of captive management in guiding management decisions, additional research into how temperament can improve behavioural management would be necessary for personality to be included in captive management plans on a systematic level. Our review aims, over a decade on from Freeman and Gosling’s initial review, to explore whether this knowledge gap has been addressed. In doing so, we aim to identify priority research directions in the hopes of facilitating effective incorporation of personality assessments into management practice in the future.

This review will adopt the following structure. We begin by describing the scope and methods of our literature survey. We then summarise the findings of that survey according to the species involved, the context (laboratory, zoo, and wild settings), the assessment methods used, and the research focus of the studies reviewed. In the following section, we discuss the findings of these studies in relation to five key applications in primate management, namely social management, environmental enrichment, training protocols, health and welfare monitoring, and conservation planning. We also consider some further directions for applied study that have received little attention to date. Finally, we conclude with some brief recommendations.

Survey Method

The selection strategy chosen for this review followed the methodology outlined in Freeman & Gosling’s (2010) more general review of personality in nonhuman primates, adapted where appropriate as outlined below. While this review primarily focuses on implications for zoo animal management, we anticipate that our findings will extend to primates in other contexts, such as laboratories and breeding facilities. Therefore, our strategy aimed to encompass a variety of disciplines, so as not to miss potentially relevant articles.

Keyword searches were conducted in a range of databases to ensure wide coverage. Databases used were ScienceDirect, Web of Science and PubMed. Databases were searched for publications which included the terms “primate”, along with “personality” or “temperament”. For this review, searches included one of the terms “management”, “conservation”, or “welfare” (e.g. “primate AND personality AND management” or “primate AND temperament AND welfare”). These terms were included to ensure results produced articles relevant to this review. As pre-2010 studies were comprehensively included in Freeman & Gosling’s (2010) review, and this review aimed to cover recent developments, searches were filtered only to include studies published in or after 2010 and until 2020. The reference lists of selected publications were also checked for additional publications. Further articles which were missed in initial searches that were brought to our attention were included where relevant.

The abstracts of articles, after eliminating duplicates, were scanned to exclude irrelevant papers. For most studies, it was straightforward to determine relevance by abstract, title and keywords. Examples of irrelevant articles included: those which did not refer to personality; studies on nonprimate taxa; nonhuman primate personality studies that did not discuss management, welfare or conservation; and studies of human personality. Furthermore, it was deemed that only peer-reviewed research articles with published empirical data would be included; thus reviews, correspondence, and conference abstracts which referred to unpublished data were excluded.

Full texts were appraised to refine the selection. Several studies which focused primarily on mechanisms underlying personality (i.e. genetic and evolutionary factors) were excluded as being beyond the scope of this study. However, several studies which measured genetic and evolutionary components were retained on the basis that those articles compared other variables against personality, such as health and welfare, and thus remained relevant. Several items were excluded as personality was discussed very minimally within the text and was not quantitatively measured.

Ultimately, 69 papers were selected for qualitative synthesis. Refer to Appendix 1 for the full list of articles.

Survey Findings

Species

While there are around 400 extant primate species, only a small proportion is represented in published personality literature. Across 69 studies, 28 species are assessed, representing 17 genera (Fig. 1)—and only nine of the 16 extant primate families. Of those 28 species, 13 were only included in one study. Within the genera represented, there is a bias towards several more commonly studied species. For example, while seven species within the genus Macaca are described across 27% of studies, rhesus macaques (Macaca mulatta) are the subjects of almost half (43%) of those articles.

Figure 1 Primate genera represented in 69 studies of primate personality published since 2010.

Some studies included assessments of multiple genera; individual genera have been counted separately.

Across all studies, 41% were carried out on just five species (Fig. 2). Chimpanzees (Pan troglodytes) were the most studied subject (16%), followed by rhesus macaques (Macaca mulatta, 12%). Of these five taxa, only two are of conservation concern (gorillas are Critically Endangered and chimpanzees are Endangered; the remaining three species are least concern: IUCN 2020). While 143 species of primate are maintained in zoos (Melfi, 2005), only 12 of these species were represented in studies of zoo-housed primates (Fig. 3). Several primate groups were represented minimally or not at all, including but not limited to lemurs, howler and spider monkeys, colobines and guenons, and gibbons, to name a few. This may be due to methodological difficulties; for example, for smaller primates such as lemurs it can be difficult to accurately identify individuals in behavioural research. Other primates, such as gibbons, are less common in captive settings and difficult to study in the wild, and so are less available as research subjects for personality studies.

Figure 2 The five most common primate species represented across 69 studies of primate personality published since 2010, compared against the proportion of studies carried out on other species.

Figure 3 Primate species represented in 69 studies of primate personality conducted in zoos.

Research setting

The highlighted studies were carried out on animals living in one of four settings. Zoo-housed animals were the most common subjects (39%), followed by laboratory (36%) and then free-living animals (25%). One study was carried out on animals in a captive open environment (1%) within a wildlife sanctuary.

Assessment method

The principal methodologies used to carry out personality assessments of nonhuman primates can be divided into three categories; (i) behavioural coding, (ii) context tests, and (iii) trait ratings (Table 1). While trait rating was the most common method used by studies, within this category there are a number of commonly used trait rating instruments. Notably, the Hominoid Personality Questionnaire (HPQ) (Weiss, 2009; 2017) was the most commonly used tool (58% of trait rating studies). Refer to Appendix 2 for a full list of trait rating instruments.

Table 1 Methods of personality assessment used in 69 studies of primate personality.

Method	Proportion of studies (%)	Definition	Example	
Context Tests	39	Subjects partake in experimental tests which are designed so that animals will react differently to a stimulus depending on their personality. E.g. Human Intruder Test, Novel Object Test	Fernández-Lázaro et al. (2019): recorded subjects’ behavioural responses to different novel objects	
Behavioural Coding	36	Observers collect behavioural data of individuals within their usual environment and code behaviours to personality	Martin & Suarez (2017): instantaneous sampling combined with event sampling to produce personality models	
Trait Rating	67	Observers who are familiar with the subject, such as a zookeeper, rates individuals against a defined set of adjectives; for example, on a scale that ranges from “absence of trait” to “displays trait frequently”.	Freeman et al. (2013): assessed personality of chimpanzees using a 41-item adjective scale	

Each method has benefits and drawbacks (Powell & Gartner, 2011; see Appendix 3). Some studies (30%) used a combination of methods to validate results and overcome some of the drawbacks associated with each process. Studies which combined behavioural coding and trait rating were more common (16%), followed by combining context tests with trait rating (9%).

Research focus

Across the reviewed personality literature, five key areas of investigation emerged. The first was method validation, while the remainder explored the role of personality in social behaviour, animal health, animal welfare, and animal management. Each study could be categorised into one of these five areas, as outlined in Table 2. Method validation was the most commonly studied single category (42%), with several studies focusing on assessing species which had not been studied before (e.g. Pritchard et al., 2014), and others on producing new assessment methods (e.g. Freeman et al., 2013; Masilkova, Weiss & Konečná, 2018).

Table 2 Research focus of 69 non-human primate personality studies published after 2010.

Topic	Percentage (%)	Percentage—Excluding Method Validation (%)	Definition	Example	
Method Validation	43	–	Studies which aimed to validate novel methods or adapt an existing method to another species	Freeman et al. (2013)	
Social Behaviour	22	39	Studies which examined relationships between personality and social behaviour	Račevska & Hill (2017)	
Animal Management	16	28	Studies which examined personality in relation to daily husbandry and management practices e.g. enrichment, training	Franks et al. (2013)	
Animal Health	10	18	Studies which examined relationships between personality and physical health, e.g. physiological measurements	Costa et al. (2020)	
Animal Welfare	9	16	Studies which examined relationships between personality and behavioural welfare indicators	Robinson et al. (2017)	

The remaining personality studies accounted for 58% of the reviewed literature. Studies of the role of personality in social behaviour were the most common within this group (22%, Table 3). These studies reported a variety of effects: a total of 27 response variables were tested against personality, of which 85% showed a statistically significant relationship, such as relationship stability (Weinstein & Capitanio, 2012) and relationship quality (Morton et al., 2015).

Table 3 Findings of personality studies on social behaviour.

Presented in order of most common species, and chronological order within species.

Species	Setting	Response variable(s)	Personality Dimension(s) Measured	Reference	
Macaca mulatta	L	Social power*	Bold(+), Excitable(+), Equable(0)	McCowan et al. (2011)	
	Intervention success*	Bold(+), Equable(+), Excitable(0)	
Macaca mulatta	L	Longitudinal relationship stability*	Equability(+), Adaptability(+), Confidence(0)	Weinstein & Capitanio (2012)	
Macaca mulatta	L	Pair success*	Emotionality(+), Nervous(+), Gentle(+)	Capitanio et al. (2017)	
Macaca mulatta	L	Trio housing success	Exploratory	Ruhde et al. (2020)	
Papio ursinus	W	Bond Strength*	Nice(+), Loner(−), Aloof(0)	Seyfarth, Silk & Cheney (2012)	
	Partner Stability*	Aloof(+), Nice(0), Loner(0)	
	Glucocorticoid levels*	Loner(+), Nice(0), Aloof(0)	
Papio ursinus	W	Problem solving*	Bold(+), Anxious(0)	Carter et al. (2014)	
	Time spent watching conspecific demonstrator	Bold(0), Anxious (0)	
	Improvement in problem solving ability after watching demonstrator*	Bold(+), Anxious(+)	
Papio ursinus	W	Bond strength*	Loner(+), Aloof(+), and Nice(+) homophily	Seyfarth, Silk & Cheney (2014)	
Macaca sylvanus	W	Social rank*	Confidence(+), Friendliness(0), Excitability(0)	Konečná et al. (2012)	
Macaca sylvanus	W	Likelihood of cooperation*	Bold(+) partners preferred, Shy(0)	Molesti & Majolo (2016)	
	Likelihood of successfully completing cooperation task	Bold(0), Shy(0)	
Pan troglodytes	Z	Rate of contact sitting within dyad*	Sociability(+), Boldness(+) and Anxious(0) homophily	Massen & Koski (2014)	
Sapajus apella	L	Affiliation within dyads*	Sociability(+), Openness(0), Neuroticism(0), Assertiveness(0), and Attentiveness(0) homophily	Morton et al. (2015)	
	Agonism within dyads*	Openness(−), Sociability(0), Neuroticism(0), Assertiveness(0), and Attentiveness(0) homophily	
	Relationship quality*	Openness(+), Sociability(+), Neuroticism(0), Assertiveness(0) and Attentiveness(0) homophily	
Callithrix jacchus	L	Group-level personality similarity*	Boldness(+), Exploration(+), Persistence (0)		
Gorilla gorilla	Z	Social resting*	Extraversion(+), Dominance(0), Fearful(0), Understanding (0)	Račevska & Hill (2017)	
	Play behaviour	Extraversion(0), Dominance(0), Fearful(0), Understanding(0)	
	Aggression*	Understanding(−), Extraversion(0), Dominance(0), Fearful(0)	
	Displacement*	Dominance(−), Extraversion(0), Fearful(0), Understanding(0)	
Macaca assamensis	W	Bond strength*	Gregarious(+), Confidence(0), Sociability(0), and Vigilance(0) homophily	Ebenau et al. (2019a)	
Pan paniscus	Z	Relationship quality*	Sociability(+), Openness(0), Boldness(0), and Activity(0) homophily	Verspeek et al. (2019)	
		Relationship compatibility*	Activity(+), Sociability(0), Openness(0), and Boldness(0) homophily		
Notes:

* Relationship with personality is statistically significant.

Z = Zoo, W = Wild free-living, L = Laboratory. + indicates significantly positive relationship, − indicates significantly negative relationship, 0 indicates no significant relationship.

The next most frequently identified category concerned personality and management practices (17%; Table 4). Across 25 response variables, 80% had a statistically significant relationship with personality, such as training success (Reamer et al., 2014; Wergård et al., 2016) and enrichment use (Franks et al., 2013; Lutz & Brown, 2018).

Table 4 Findings of personality studies on animal management.

Presented in order of most common species, and chronological order within species.

Species	Setting	Response variable(s)	Personality Dimension(s) Measured	Reference	
Pan troglodytes	Z	Training participation*	Openness(+), Dominance(0), Conscientious(0), Agreeableness(0), Extraversion(0), Neuroticism(0)	Herrelko, Vick & Buchanan-Smith (2012)	
	Self-directed behaviour frequency during research*	Conscientious(+), Neuroticism(+), Openness(0), Dominance(0), Agreeableness(0), Extraversion(0)	
	Vigilance during research*	Neuroticism(+), Openness(0), Dominance(0), Conscientious(0), Agreeableness(0), Extraversion(0)	
Pan troglodytes	L	Problem-solving success*	Dominance(+), Methodical(+), Reactivity(0), Openness(0), Agreeableness(0), Extraversion(0)	Hopper et al. (2014)	
Pan troglodytes	L	Initial training participation*	Openness(+), Agreeableness(0), Conscientious(0), Reactivity(0), Dominance(0), Extraversion(0)	Reamer et al. (2014)	
	Level of participation	Openness(0), Agreeableness(0), Conscientious(0), Reactivity(0), Dominance(0), Extraversion(0)	
Pan troglodytes	Z	Participation* in research task	Dominance(+), Conscientious(+), Openness(+), Neuroticism(-), Agreeableness(0)	Altschul et al. (2017)	
		Dropout rate*	Conscientious(+), Agreeableness(+), Dominance(0), Openness(0), Neuroticism(0)	
	Task accuracy*	Extraversion(+), Conscientious(0), Agreeableness(0), Dominance(0), Openness(0), Neuroticism(0)	
	Engagement in research task*	Openness(+), Agreeableness(-), Extraversion(0), Dominance(0), Conscientious(0), Neuroticism(0)	
Macaca fascicularis	L	Training success*	Activity(+), Emotionality(0), Sociability(0), Tolerance(0)	Wergård et al. (2016)	
Macaca fascicularis	L	Porch enrichment usage*	Bold x porch location(+)	Lutz & Brown (2018)	
Saimiri sciureus	Z	Viewing window approach*	Playfulness(+), Cautious(-), Solitary(-), Depressed(-), Dominance(0), Affectionate(0), Friendly(0), Gentle(0)	Polgár, Wood & Haskell (2017)	
		Research participation*	Playfulness(+), Affectionate(+), Friendly(+), Gentle(+), Cautious(-), Dominance(0), Solitary(0), Depressed(0)	
Gorilla gorilla	Z	Activity budgets under high/low crowd conditions	Extraversion(0), Dominant(0), Fearful(0), Understanding(0)	Stoinski, Jaicks & Drayton (2012)	
Saguinus oedipus	Z	Speed of enrichment approach*	Promotion-focused(+)	Franks et al. (2013)	
	Vigilance towards unfamiliar enrichment*	Prevention-focused(+),	
Sapajus apella	Z	Test participation*	Openness(+), Assertiveness(0), Neuroticism(0), Sociability(0), Attentiveness(0)	Morton et al. (2013)	
		Test performance*	Openness(+), Assertiveness(-), Neuroticism(0), Sociability(0), Attentiveness(0)		
Macaca nigra	Z	Training cue-response latency	Boldness(0), Adaptability(0), Fearfulness(0)	Ward & Melfi (2013)	
Macaca mulatta	L	Rewarded trials*	Friendliness(+), Openness(0), Anxiety(0), Activity(0), Dominance(0), Confidence(0)	Altschul, Terrace & Weiss (2016)	
Progress*	Friendliness(+), Openness(+),Anxiety(0), Activity(0), Dominance(0), Confidence(0)	
Error*	Friendliness(-), Openness(-), Anxiety(0), Activity(0), Dominance(0), Confidence(0)	
Reaction time	Friendliness(0), Openness(0), Anxiety(0), Activity(0), Dominance(0), Confidence(0)	
Notes:

* Relationship with personality is statistically significant.

Z = Zoo, W = Wild free-living, L = Laboratory. + indicates significantly positive relationship, − indicates significantly negative relationship, 0 indicates no significant relationship.

In the final two categories, welfare (10%; Table 5) and health (9%; Table 6), the studies primarily addressed well-being concerns, such as stress. Welfare studies primarily examined behavioural indicators of stress such as stereotypies (Vandeleest, McCowan & Capitanio, 2011; Robinson et al., 2016; Robinson et al., 2017), while health studies examined stress-related morbidity risk (Fernandez & Timberlake, 2008; Gottlieb et al., 2018; Robinson et al., 2018). Notably, only one study examined personality and health status in a wild population (Costa et al., 2020).

Table 5 Findings of personality studies on animal health.

The most common species are presented first, and chronological order within species.

Species	Setting	Response variable	Personality Dimension(s) Measured	Reference	
Macaca mulatta	L	Motor stereotypy risk	Gentle(0), Nervous(0)	Vandeleest, McCowan & Capitanio (2011)	
Macaca mulatta	L	Motor stereotype development*	Gentle(−), Vigilant(0), Confident(0), Nervous(0)	Gottlieb, Capitanio & McCowan (2013)	
Gorilla gorilla	Z	Subjective well-being*	Extraversion/Agreeableness(+), Dominance(+), Conscientious(0)	Schaefer & Steklis (2014)	
Sapajus apella	L	Subjective well-being*	Assertiveness(+), Sociability(+), Openness(0), Neuroticism(0), Attentiveness(0)	Robinson et al. (2016)	
Pan troglodytes	Z	Subjective well-being*	Openness(+), Extraversion(+), Neuroticism(−), Dominance(0), Agreeableness(0), Conscientious(0)	Robinson et al. (2017)	
Callithrix jacchus	L	Subjective well-being*	Sociability(+), Neuroticism(−), Dominance(0)	Inoue-Murayama et al. (2018)	
	Hair cortisol levels*	Dominance(+), Sociability(+), Neuroticism(0)	
Notes:

* Relationship with personality is statistically significant.

Z = Zoo, W = Wild free-living, L = Laboratory. + indicates significantly positive relationship, − indicates significantly negative relationship, 0 indicates no significant relationship.

Table 6 Findings of personality studies on animal welfare. Presented in chronological order.

Species	Setting	Response variable	Personality Dimension(s) Measured	Reference	
Macaca mulatta	L	Diarrhea incidence	Vigilant(0), Gentle(0), Nervous(0), Confident(0)	Gottlieb et al. (2018)	
	Risk of chronic diarrhea with relocations*	Confident(+), Gentle(−), Nervous(−), Vigilant(0)	
Macaca mulatta	L	Number of injuries*	Confidence(−), Anxiety(−), Dominance(0), Openness(0)	Robinson et al. (2018)	
	Number of illnesses	Confidence(0), Anxiety(0), Dominance(0), Openness(0)	
Rhinopithecus roxellana	Z	Number of illnesses*	Aggressiveness(−), Sociability(0), Excitability(0), Mellowness(0)	Jin et al. (2013)	
	Illness duration*	Aggressiveness(−), Sociability(0), Excitability(0), Mellowness(0)	
	Gastrointestinal function*	Sociability(+ in young individuals,—in older individuals). Excitability(+), Aggressiveness(−), Mellowness(−)	
Gorilla gorilla	Z	Lifespan*	Extraversion(+), Dominance(0), Neuroticism(0), Agreeableness(0)	Weiss et al. (2013)	
Pan troglodytes	Z	Lifespan*	Agreeableness(+), Extraversion(0), Conscientious(0), Openness(0), Neuroticism(0)	Altschul et al. (2018)	
Sagiunus oedipus
Saguinus imperator
Leontopithecus rosalia
Callithrix jacchus
Callithris geoffroyi
Cebuella pygmaea
Pithecia pithecia	Z	Fecal cortisol*	Active(+), Aggressive(+), Playful(+), Lazy(0), Subordinate(0), Fearful(0)	Fernández-Lázaro et al. (2019)	
Leontopithecus chrysomelas	W	Time spent foraging*	Confidence(+)	Costa et al. (2020)	
	Body mass*	Confidence(+)	
	Respiratory frequency	Confidence(0)	
	Body Mass Index	Confidence(0)	
	Heart Rate	Confidence(0)	
	Body temperature	Confidence(0)	
	FGM levels	Confidence(0)	
Notes:

* Relationship with personality is statistically significant.

Z = Zoo, W = Wild free-living, L = Laboratory. + indicates significantly positive relationship, − indicates significantly negative relationship, 0 indicates no significant relationship.

Applications to Primate Management

While rudimentary descriptions of animal personality traits have long been informally recognised (Watters & Powell, 2012), the past decade has seen increasing scientific recognition that consistent individual differences exist, are measurable, and impact individual and population-level management outcomes. However, this review identified that research to date continues to focus on the validation of new assessment methods and, by comparison, researchers have explored the links between personality and management outcomes in less detail. It is also notable that the taxonomic coverage in the field is biased towards well-studied primates which already have validated assessments, while research on other taxa—such as lemurs, gibbons, and howler monkeys, to name a few—is limited to descriptive studies or non-existent. This is despite the potential benefits of understanding personality in the management of species which are commonly found in collections or are of high conservation priority. Furthermore, the potential of personality assessments as a management tool is still in need of exploration before they can be widely applied.

We now extend our review to the discussion of the key findings of the identified personality studies, with specific focus on how the knowledge base could be incorporated into primate management practices. Through this review, we identified five major research focuses: method validation, social behaviour, management, welfare and health. However, as each category is broad, covering a variety of possible applications, for the purposes of this discussion it was elected to highlight more specific areas of management practice which the identified studies could inform. The following five key areas emerged, namely environmental enrichment, training protocols, health and welfare monitoring, and conservation planning.

Social management

The link between social behaviour and personality has been a recent focus of several studies (Tables 2 and 3). The development and maintenance of social bonds are essential aspects of primate behaviour, and most species form social groups in the wild (Lehmann, Korstjens & Dunbar, 2007). Housing primates in social groups which mimic those of their wild conspecifics confers considerable benefits: promoting naturalistic behaviour budgets, providing mental stimulation, and reducing stress (Kleiman, Thompson & Baer, 2012); however, it also comes with risks. For example, species which would normally live in multimale groups in the wild are generally exhibited in single-male groups for breeding purposes and to manage aggression (Schapiro, 2017). Furthermore, facilities often need to move individuals between breeding groups; social disruptions, when unsuccessful, can result in physical injury and compromised psychological health (Kleiman, Thompson & Baer, 2012; Brando & Buchanan-Smith, 2018). Coleman (2012) proposed that personality assessments may be used by managers to guide socialization choices in captive primates.

The risk of injury due to conspecific aggression is a key consideration for primate managers, so an ability to predict the risk of aggression from personality could be very useful. In a recent study, Robinson et al. (2018) found that rhesus macaques, which were housed in recently established social groups, which rated as either low in Confidence or low in Anxiety presented with significantly more injuries. Personality was associated with the number of injuries even when controlling for kinship, rank, and sex. These results seem to indicate that individuals higher or lower in particular personality traits are at higher risk of experiencing aggression when placed in new groups. Alternatively, personality could predict the individuals more likely to instigate aggression. The relationship between personality and aggressive and antisocial behaviour is well-described in the human literature; particularly, low Agreeableness, low Conscientiousness and high Neuroticism contribute to aggression in studies of humans (Jones, Miller & Lynam, 2011). However, too few studies systematically address the relationship between personality and aggressive behaviour to form a meaningful conclusion in nonhuman primates; while low ‘Understanding’ was associated with higher aggression in one bachelor group of four gorillas (Račevska & Hill, 2017), the sample size was too small to control for factors such as social rank. While further studies into nonhuman primates would be necessary to confirm the relationship between personality and aggression, the current findings on aggression and injury in both humans and primates could have potential implications for social management decisions. For example, individuals higher or lower in traits related to aggression may require different introduction protocols to reduce the likelihood of injury. It would be interesting for future research to examine whether the same personality dimensions which influence aggressive behaviour in humans are of importance in predicting aggression in nonhuman primates.

Conversely from aggression, recent research suggests that certain personality dimensions may play an important role in the formation and maintenance of primate social bonds. One study of rhesus macaques found that personality data collected up to 10 years prior was associated with later pair success, with successful pairs being those which actively sought each other’s company and did not show fear or aggression outside of feeding (Capitanio et al., 2017). These results suggest that personality data collected by keepers could be used long-term to aid in forming new social groups by matching personality types. A “Sociability” dimension has been described in multiple primate species (Freeman & Gosling, 2010; Freeman et al., 2013; Morton, Lee & Buchanan-Smith, 2013; Pritchard et al., 2014; Martin & Suarez, 2017). Primate Sociability is associated with adjectives such as “Helpful” and “Sociable” (King & Figueredo, 1997) and corresponds with a range of behaviours including proximity, grooming, and play (Koski, 2011). A study by Morton et al. (2015) found that captive capuchin (Sapajus spp.) dyads who were more similar in the Sociability dimension, had higher-quality relationships. Similar findings in bonobos (Verspeek et al., 2019), chimpanzees (Massen & Koski, 2014) and Assamese macaques (Ebenau et al., 2019a) suggest that personality dimensions related to social behaviour may play an important role in the development of stronger social bonds. Overall, these results indicate that individuals which are more sociable seem to demonstrate a preference for animals which are similarly sociable; at the other end of the scale, individuals which are less sociable generally appear to prefer being with similarly less sociable conspecifics.

In addition to individual dyadic relationships, individual personalities will also influence the social network (Seyfarth, Silk & Cheney, 2014). For example, research by McCowan et al. (2011) found that high-ranking rhesus macaque males rated as higher in Equable were more successful interveners; and that successful third-party intervention resulted in more stable social networks, less wounding, and higher rates of post-conflict reconciliation. However, social networks are also influenced by rank and kinship (Seyfarth, Silk & Cheney, 2012; Seyfarth, Silk & Cheney, 2014). One study of laboratory-housed rhesus macaques identified sex and relatedness as being just as essential predictors of relationship stability as personality homophily (Weinstein & Capitanio, 2012). The importance of relatedness is corroborated by research on wild female chacma baboons, where personality homophily was only significant in predicting the relationship strength of closely related siblings; personality was not a predictor of relationship quality in unrelated pairs, where similarities in age and rank were more significant (Seyfarth, Silk & Cheney, 2014). Consequently, the extent to which personality would aid management, particularly for large groups and unrelated individuals, is called into question. It would be useful for future research to examine whether personality variables impact the success with which new individuals are introduced into captive social groups. While experimentally manipulating social groups of primates with limited populations, especially endangered species, is not possible in many cases, retroactive examination of group success for animals which can have their personalities assessed could still be instructive.

Overall, these results suggest that while various aspects of an individual’s personality may be used to guide social management decisions in primates, caution should be exercised to avoid over-simplified generalisations. While personality data provides additional information to consider during social management and serves as an additional tool in predicting the outcome of decisions, it remains imperative, and more feasible, to consider all characteristics of the individuals involved including age, sex, rank, and relatedness.

Environmental enrichment

Environmental enrichment is a crucial component of many animal husbandry programmes. Enrichment promotes welfare by encouraging species-typical behaviour, and challenging animals both mentally and physically (Gartner & Weiss, 2018) and comes in several forms, providing unique opportunities to forage, explore and manipulate their environment (Hosey, Melfi & Pankhurst, 2013). However, enrichment does not consistently improve welfare for all individuals; one study of common squirrel monkeys (Saimiri sciureus) identified consistent individual differences in the extent of welfare improvement under different enrichment conditions, which were unexplained by sex or age (Izzo, Bashaw & Campbell, 2011). A possible explanation for these consistent individual differences is that they are reflections of personality. Correlations between personality and behaviour towards enrichment objects have been recently described in snow leopards (Panthera unca) (Gartner & Powell, 2012), which suggests that accounting for personality could, in theory, improve enrichment efficiency. However, the link between personality and enrichment success is understudied in primates. To date, only one paper, published prior to 2010 and therefore not included in our analysis, provides evidence that enrichment plans based on personality have tangible welfare benefits for primates (Highfill, 2008). Studying a small group of Garnett’s bushbabies (Otolemur garnettii), Highfill found that enrichment interventions designed to reflect personality dimensions resulted in significant reductions in stereotypic behaviour.

The limited research on personality and enrichment across primate and non-primate taxa is restricted to restrictive comparisons of just one or two personality dimensions and small subsets of behaviours. While novel objects are often used in assessments designed to measure certain personality components, such as the “Shy—Bold” continuum (Verdolin & Harper, 2013; Šlipogor et al., 2016; Fernández-Lázaro et al., 2019), the effect is rarely described in the context of enrichment effectiveness. Furthermore, while associations between “Shy—Bold” and novel object approach are described, the effectiveness of other common enrichment interventions—such as scatter feeding, training, puzzle toys etc.—may rely on different dimensions of personality (Highfill, 2008). Consequently, studying the interaction between personality and enrichment may be difficult to apply across settings. Franks et al. (2013) suggest that animals display an individual preference for “promotion” (rewards-motivated) or “prevention” (safety-motivated) based on the “regulatory focus” personality theory (Higgins, 1997). In their study, the authors compared behaviour-coded regulatory focus personalities against latency to approach different enrichment items for four zoo-housed cotton-top tamarins (Saguinus oedipus) and found significant correlations between personality and enrichment approach. While the small sample size limits how far the results can be generalised, the results demonstrate how personality can be used to successfully predict how individuals approach enrichment in a small collection. However, the relationship between personality and enrichment intervention success remains unclear. It would be beneficial for researchers to further explore the impact of personality on enrichment success, including whether the results demonstrated by Highfill (2008) can be replicated in other primate species. Personality assessments could then be used to produce enrichment protocols which are tailored to the individuals.

Training protocols

Training has a multitude of benefits and is a staple component of many primate management programmes. Training provides mental stimulation and thus is a form of enrichment (Melfi, Dorey & Ward, 2020), and animals can voluntarily participate in common management procedures, such as veterinary examinations, without the need for more invasive methods of restraint (Reamer et al., 2014). However, while some animals are eager to participate and learn quickly, others can struggle to pick up the same tasks or are more reluctant to engage in training (Melfi, Dorey & Ward, 2020). Two studies have found that Openness predicts participation in training sessions in chimpanzees (Herrelko, Vick & Buchanan-Smith, 2012; Reamer et al., 2014; Altschul et al., 2017), and similar findings have been reported in other species (Altschul, Terrace & Weiss, 2016). A recent study of long-tailed macaques (Macaca fascicularis) demonstrated that high scores on Activity, a dimension similar to chimpanzee Openness, correlated with a decrease in the number of approximations required to successfully train new behaviours (Wergård et al., 2016). In another study, squirrel monkeys rated as low on “Cautiousness” and high on “Playful”, “Gentle”, “Affectionate” and “Friendly” were more likely to participate in research procedures (Polgár, Wood & Haskell, 2017). These results have implications for training programmes; personality data could be used to identify which individuals to target for training first versus those which may require additional support or alternative training methods.

In their review of great ape personality research, Gartner & Weiss (2018) suggest that more confident animals may act as positive role models for those which are shyer and more reluctant to train. It may therefore be beneficial to examine the effect of personality on cue-response latency, or the time it takes for a target animal to complete a desired behaviour upon being given a verbal or visual command by a trainer, to determine which individuals are best suited as conspecific trainers for other individuals. In a multi-institution and cross-species study, Ward & Melfi (2013) measured cue-response latency in 12 Sulawesi crested macaques (Macaca nigra), eight black rhinoceros (Diceros bicornis), and 11 Chapman’s zebra (Equus burchellii) and compared results against keeper-rated behavioural profiles. While the authors highlight that the fastest response times were from “Bold” dominant monkeys in two of the three macaque groups, the pattern identified is anecdotal and the relationship between personality and latency was not found to be significant. However, only three groups of each species were assessed, and groups varied in their level of past training experience from completely untrained to fully trained. When compared with nonprimate taxa, social species responded significantly faster than solitary species (Ward & Melfi, 2013), which may imply that social learning facilitates better performance during training. Directed social learning, where individuals preferentially learn from specific demonstrators, has been described in captive chimpanzees (Kendal et al., 2015) and wild vervet monkeys (Chlorocebus pygerythrus) (Grampp et al., 2019; Canteloup, Hoppitt & van de Waal, 2020), which are described with a preference for observing dominant animals. However, it is yet unexplored whether primates demonstrate a preference for particular personalities in demonstrators, despite evidence suggesting that personality may play a role in social learning. A recent study of cooperation in free-living Barbary macaques (Macaca sylvanus) noted that “Shy” subjects, while less likely to approach a novel test apparatus, maintained cooperation for longer when working with bold individuals (Molesti & Majolo, 2016). Carter et al. (2014) found that chacma baboons rated as “Anxious” showed greater improvement in completing problem-solving tasks after observing experienced conspecific demonstrators. While past research has primarily focused on training and personality at the individual level, in light of these findings, future studies could explore the group-level impacts of personality and whether the presence of bold partners significantly influences training success of shyer primates.

Health and welfare monitoring

The maintenance of physically and mentally healthy animals is a key aspect of primate management. Studies have found links between personality and these areas (Tables 5 and 6); for example, personality has been linked to longevity in captive primate species such as chimpanzees (Altschul et al., 2018) and gorillas (Weiss et al., 2013). However, there is little evidence to suggest personality alone predisposes individuals to specific health outcomes (Jin et al., 2013; Robinson et al., 2018); instead, the recent literature suggests personality influences health when interacting with factors such as environment and stress (Gottlieb et al., 2018). This relationship would suggest that when faced with stressful events, such as relocating enclosures, specific individuals may have a lower stress tolerance threshold and thus are more vulnerable to stress-related illness. For example, a recent study by Gottlieb et al. (2018) found that, while personality did not predict an individual’s risk of acute diarrhoea after a housing relocation, more “Nervous” and “Gentle” monkeys were more likely to experience chronic diarrhoea. An additional study of golden snub-nosed monkeys (Rhinopithecus roxellana), an endangered species at high risk of gastrointestinal disorders due to their specialist diet, found a significant relationship between personality and morbidity. Lower Aggressiveness was related to greater incidence of illness, longer illness duration, and poorer digestive function (Jin et al., 2013). Understanding how personality influences stress has implications for primate welfare and health management; for example, it may be possible to promote longevity through understanding specific animals’ limits for stress and identifying individuals at high risk of developing stress-related illnesses. Resources can then be allocated to focus preventative measures on individuals predisposed to poor health outcomes, and potentially stressful management interventions can be avoided or minimised for those individuals.

Monitoring the mental health of animals for indicators of stress can present challenges to animal caretakers. Welfare is, typically, measured with one or a combination of methods; behavioural indicators, such as stereotyped behaviour, and physiological measures, such as cortisol, are commonly utilised indices (Hosey, Melfi & Pankhurst, 2013). However, correlations between these variables are not always consistent, even within the same species (Fernández-Lázaro et al., 2019); consequently, animal caretakers may inadvertently over- or under-estimate welfare. Studying personality may aid in understanding why behavioural and physiological measures are not always consistent measures of welfare; for example, recent studies have suggested that individuals may rely on different coping strategies and display alternative stress-indicative behaviours (Ferreira et al., 2018). However, while several studies have highlighted personality as an intrinsic factor predisposing individuals to developing stereotyped behaviours, they typically focus on only one type of behaviour. Recent research by Robinson et al. (2016, 2017) which found associations between primate personality and psychological health, for example, relied on motor stereotypies to generate welfare scores. Similarly, while Gottlieb, Capitanio & McCowan (2013) identified a relationship between life history and personality in predicting motor stereotyped behaviour (i.e., pacing) and self-injurious behaviour risk, the authors acknowledge that the absence of such behaviours does not necessarily indicate “positive” welfare. Indeed, for some animals, inactivity or unresponsiveness could be less preferable than pacing from a welfare perspective, and yet in the aforementioned studies may have scored as having higher welfare.

Only one study explored how traditional welfare measures may be impacted by personality. Ferreira et al. (2018) aimed to quantitively measure the hormonal correlates of personality types and specific stress-indicative behaviours under stress in 25 zoo-housed brown capuchins. The “Active” personality dimension was found to be of particular significance in this study; monkeys assessed as more “Active” displayed more rapid stereotyped behaviours, such as pacing head-twirling, and exhibited higher faecal glucocorticoid metabolite (FGM) levels (Ferreira et al., 2018). “Active” was similarly reported to predict FGM in an assessment carried out by Fernández-Lázaro et al. (2019) on eight primate species (S. oedipus, Saguinus imperator, Leontopithecus rosalia, Callithrix jacchus, Callithrix geoffroyi, Cebuella pygmaea, Pithecia pithecia, Nycticebus pygmaeus). In contrast, less “Active” animals were more likely to display prolonged state stereotyped behaviours, such as self-scratching and inertia (Ferreira et al., 2018). Understanding individual variation in stress coping styles would aid in explaining the incongruity between measures of welfare and would support primate caretakers in recognising signs of poor welfare in their animals.

Robinson et al. (2017) suggest that a subjective approach to measuring welfare, in combination with traditional measurements such as stress-indicative behaviours and hormone levels, provides a quick and easy method to measuring welfare which considers personality. Comparisons of caretaker-rated subjective well-being scores and personality have highlighted dimensions of potential relevance. High Neuroticism was a predictor of low welfare in studies of chimpanzees (Robinson et al., 2017), capuchins (Robinson et al., 2016) and marmosets (Inoue-Murayama et al., 2018), while high Dominance was related to better welfare in male gorillas (Schaefer & Steklis, 2014). Equipped with the knowledge that particular personality dimensions influence the risk of poor welfare, and that certain indicators may be more relevant for particular individuals, primate keepers can identify at-risk animals and monitor them more closely. It would be particularly beneficial to establish individual behavioural baselines so that dramatic changes unusual for a specific animal are easy to identify.

Conservation planning

For endangered species which are bred in captivity, conservation plans may consider eventual reintroduction of captive populations into the wild, or translocation of free-living populations to new areas. Hosey, Melfi & Pankhurst (2013) highlight how personality is, increasingly, coming to be appreciated for its contribution to the success of such initiatives. There is rising concern that captive breeding programmes which ignore personality are inadvertently reducing the diversity of behavioural traits which contribute to survival (McDougall et al., 2006). An area of literature suggests that personality has a genetic component (e.g. Adams, King & Weiss, 2012; Inoue-Murayama et al., 2018) and up to 50% of personality variation, depending on the species, is heritable (van Oers et al., 2005; Dochtermann, Schwab & Sih, 2015). Therefore, breeding to select for high boldness may be beneficial for captive populations as bold animals may be less likely to suffer adverse reactions to the presence of human caretakers and visitors (Stoinski, Jaicks & Drayton, 2012; Verdolin & Harper, 2013; Polgár, Wood & Haskell, 2017). In the wild, however, bolder animals may be more likely to engage in risky behaviour, such as approaching unfamiliar humans or predators, which could compromise the likelihood of survival (Dammhahn & Almeling, 2012).

Personality may impact survivorship and fitness of wild populations in several ways. For example, there is evidence to suggest that personality influences antipredator responses in free-living primates. Blaszczyk (2017) assessed whether experimental assays of Boldness could predict antipredator response to both natural and artificial novel predators in wild vervet monkeys. In this study, a Boldness score generated after three novel object tests correlated with exploratory risk-taking behaviour when faced with a predator stimulus; bolder monkeys were more likely to approach and inspect both novel and natural predators (Blaszczyk, 2017). These results suggest that certain individuals may be less suitable to reintroductions based on their personality. While there has yet to be a quantitative assessment of personality and reintroduction success of primates, a study of captive-bred swift foxes (Vulpes velox) found that individuals assessed as Bold had decreased likelihood of survival in the first 6 months post-release (Bremner-Harrison, Prodohl & Elwood, 2004). Selecting candidates for release based on their personality should, therefore, increase the success of reintroductions.

However, taking a one-size fits all approach for applying personality to reintroductions may prove to be too simplistic. As Boldness is naturally present in wild populations, it must be associated with benefits to have evolved. It is possible that removing the dimension in reintroduced groups—while potentially improving initial survivorship—could have unforeseen impacts on other aspects of wild living. For example, bolder animals may be more inclined to explore foraging areas for new food patches; for instance, Costa et al. (2020) found that in wild golden-headed lion tamarins (Leontopithecus chrysomelas) high “Confidence” was significantly correlated with more time spent foraging and higher body mass. Furthermore, Carter et al. (2012) argued that the traditional methods used for measuring Boldness in wild populations, such as predator tests and novel food tests, inconsistently measure the personality dimensions relevant for survival. The authors recorded individual responses to both a novel predator test and a novel food test for 57 wild baboons and examined correlations between the results of both assays. Interestingly, the animals which showed the greatest alarm response to the novel predator—and thus scored as low in Boldness—inspected the item for longer and scored higher on Boldness in the food test (Carter et al., 2012). Consequently, it was determined that one of the assays must have been measuring a different personality dimension than Boldness. It should be recognised that multiple personality dimensions contribute to the long-term success of primate reintroductions. It would be beneficial to carefully monitor reintroductions for animals of a range of personality types to fully assess which dimensions are associated with positive outcomes.

Further Directions for Applied Study

There are several further areas of interest where personality could be used to guide and improve primate management but have received little or no attention to date. The first is the relationship between personality and reproductive success. Seyfarth, Silk & Cheney (2012) introduced this concept by suggesting that personality is connected to social measures which influence reproductive success in wild baboons, however, the topic was not extensively discussed in the reviewed literature otherwise. The relationship between personality and reproductive success has nonetheless been documented in several species, including humans (Berg et al., 2014). For example, a 1999 study of captive cheetahs found that individual behavioural differences were associated with breeding status (Wielebnowski, 1999). A similar relationship was identified in captive black rhinoceros (Carlstead, Mellen & Kleiman, 1999). These findings suggest that certain personality traits are correlated with higher reproductive success in captivity, information which would be beneficial to breeding programmes of species which have historically been difficult to breed in the zoo environment. Furthermore, Powell & Gartner (2011) suggested that personality may play a role in mate selection. A relationship between personality and mate compatibility has been described in giant pandas (Martin-Wintle et al., 2017) but no research has yet examined this relationship in nonhuman primates. As information regarding the reproductive success of primates in captivity is readily available through studbook coordinators and similar avenues, comparisons between personality and reproductive status, pair success, fertility and other variables would be straightforward for future studies to explore.

Second, personality could be used to decide which animals are most suitable for visitor education experiences in zoos, such as walk-through exhibits or meet-and-greets. While this possibility has not been quantitatively explored, Polgár, Wood & Haskell (2017) found that more Playful and less Solitary and Cautious squirrel monkeys were significantly more likely to approach a visitor viewing window under high crowd conditions, which suggests that certain personality types may be more comfortable with unfamiliar human presence. This effect might also extend to the human keepers of animals and human-animal interactions in general.

Conclusions

Accumulating evidence suggests that there can be important links between an individual’s personality and its social behaviour, management, welfare and health, and that incorporating these links into captive breeding and conservation programmes could lead to more successful and positive outcomes. On this basis, we make the following recommendations: Research focus within the field of primate personality research should shift away from the development and validation of more personality tools and instead towards (i) standardising common tools, such as the Hominoid Personality Questionnaire, for use across taxa, and (ii) making such standardised tools easier to implement on a wide scale. It would be beneficial to assess whether personality assessments validated at the family level are adequate for use across all genera and species within that family. These assessments will also need to be validated for use in understudied primate groups—including, but not limited to, tarsiers, lemurs, howler monkeys, night monkeys, langurs, and gibbons—to facilitate the following recommendations for a wider range of species.

Wider species management initiatives (e.g. EAZA ex-situ programmes) should encourage participating facilities to carry out personality assessments. Larger samples of personality data collected across multiple institutions could prove a valuable resource for co-ordinated breeding programmes, as well as providing the scope for longitudinal and retrospective studies.

Future study should address key gaps in the primate personality literature; particularly (i) regarding taxa which are currently underrepresented in studies and (ii) exploring the links between personality and health, welfare, social management, and other practical areas of interest in greater detail. Further areas of interest include the relationship between personality and reproductive success and human-animal interactions.

We are grateful to Louise Barrett, Alexander Weiss, and an anonymous reviewer for their very helpful comments on this paper.

Appendix 1 Table of surveyed data for 69 published articles on primate personality which were highlighted for this review, presented in order of publication.

Study	Species	Context	Focus	Method	
Highfill et al. (2010)	Garnett’s Bushbaby
(Otolemur garnettii)	Zoo-housed animals	Method Validation—comparing rating and coding as assessment methods	Trait Rating + Behaviour Coding	
McCowan et al. (2011)	Rhesus Macaque
(Macaca mulatta)	Laboratory animals	Social Behaviour—demonstrating role of personality in social network dynamics	Trait Rating + Behaviour Coding	
Koski (2011)	Chimpanzee
(Pan troglodytes)	Zoo-housed animals	Method Validation—examining predictive validity of social personality traits over time	Behaviour Coding	
Vandeleest, McCowan & Capitanio (2011)	Rhesus Macaque
(Macaca mulatta)	Laboratory animals	Animal Welfare—relationship between personality and motor stereotypy risk	Context Tests + Trait Rating	
Carter et al. (2012)	Chacma Baboon
(Papio ursinus)	Free-living animals	Method Validation—testing cross-context consistency of a personality assessment	Context Tests	
Dammhahn & Almeling (2012)	Mouse Lemur
(Microcebus murinus)	Free-living animals	Method Validation—testing cross-context consistency of a personality assessment	Context Tests	
Herrelko, Vick & Buchanan-Smith (2012)	Chimpanzee
(Pan troglodytes)	Zoo-housed animals	Management—influence of personality on behaviour outcomes during training	Trait Rating	
Konečná et al. (2012)	Barbary Macaque
(Macaca sylvanus)	Free-living animals	Social Behaviour—examining relationship between personality and social rank	Trait Rating	
Seyfarth, Silk & Cheney (2012)	Chacma Baboon
(Papio ursinus)	Free-living animals	Social Behaviour—examining correlations between personality and fitness	Behaviour Coding	
Stoinski, Jaicks & Drayton (2012)	Gorilla
(Gorilla gorilla)	Zoo-housed animals	Management—comparing individual differences and response to visitor numbers	Trait Rating	
Weinstein & Capitanio (2012)	Rhesus Macaque
(Macaca mulatta)	Laboratory animals	Social Behaviour—longitudinal study of personality and friendship stability	Context Tests + Trait Rating	
Franks et al. (2013)	Cotton-top Tamarin
(Saguinus oedipus)	Zoo-housed animals	Management—examining relationship between personality and enrichment use	Context Tests	
Freeman et al. (2013)	Chimpanzee
(Pan troglodytes)	Laboratory animals	Method Validation—assessing validity of novel, comprehensive assessment	Trait Rating + Behaviour Coding	
Gottlieb, Capitanio & McCowan (2013)	Rhesus Macaque
(Macaca mulatta)	Laboratory animals	Animal Welfare—relationship between personality and stereotypic behaviours	Context Tests + Trait Rating	
Iwanicki & Lehmann (2015)	Common Marmoset (Callithrix jacchus)	Zoo-housed animals	Method Validation—reliability and validity of personality in marmosets	Trait Rating + Behaviour Coding	
Jin et al. (2013)	Golden snub-nosed monkey
(Rhinopithecus roxellana)	Zoo-housed animals	Animal Health—relationship between personality and gastrointestinal health	Trait Rating	
Manson & Perry (2013)	White-faced Capuchin (Cebus capucinus)	Free-living animals	Method Validation—Structure, sex differences and temporal stability of personality	Trait Rating	
Massen et al. (2013)	Chimpanzee (Pan troglodytes)	Zoo-housed animals	Method Validation—repeatability and consistency of personality traits over time	Context Tests	
Morton, Lee & Buchanan-Smith (2013)	Brown Capuchin
(Sapajus apella)	Laboratory animals	Method Validation—comparing personality structure with other primate species	Trait Rating + Behaviour Coding	
Morton, Lee & Buchanan-Smith (2013)	Brown Capuchin
(Sapajus apella)	Zoo-housed animals	Management—personality differences in training participation	Trait Rating	
Uher, Addessi & Visalberghi (2013)	Cynomulgus macaque (Macaca fascilaris)	Laboratory Animals	Method Validation—representations of personality from experienced and novice raters	Behaviour Coding + Trait Rating	
Uher, Werner & Gosselt (2013)	Brown capuchin (Sapajus apella)	Laboratory Animals	Method Validation—differences between context tests and behavioural coding	Behaviour Coding + Context Tests	
Verdolin & Harper (2013)	Mouse Lemur
(Microcebus murinus)	Zoo-housed animals	Method Validation—demonstrating consistency in personality across contexts	Context Tests	
Ward & Melfi (2013)	Sulawesi Crested Macaque
(Macaca nigra)	Zoo-housed animals	Management—comparing personality profiles with training response latency	Trait Rating	
Weiss et al. (2013)	Gorilla
(Gorilla gorilla)	Zoo-housed animals	Animal Health—examining the relationship between personality and lifespan	Trait Rating	
Carter et al. (2014)	Chacma Baboon
(Papio ursinus)	Free-living animals	Social Behaviour—associations between personality and social learning	Context Tests	
Hopper et al. (2014)	Chimpanzee
(Pan troglodytes)	Laboratory animals	Management—influence of internal factors on problem-solving success	Trait Rating	
Pritchard et al. (2014)	Tibetan Macaque
(Macaca thibetana)	Free-living animals	Method Validation—testing methodology for personality assessment in Tibetan macaques	Behaviour Coding + Trait Rating	
Massen & Koski (2014)	Chimpanzee
(Pan troglodytes)	Zoo-housed animals	Social Behaviour—examining relationship between personality and social behaviour	Behaviour Coding + Context Tests	
Reamer et al. (2014)	Chimpanzee
(Pan troglodytes)	Laboratory animals	Animal Management—internal factors which may impact initial training success	Behaviour Coding + Trait Rating	
Schaefer & Steklis (2014)	Gorilla
(Gorilla gorilla)	Zoo-housed animals	Animal Welfare—personality and subjective well-being in gorilla bachelor groups	Trait Rating	
Seyfarth, Silk & Cheney (2014)	Chacma Baboon
(Papio Ursinus)	Free-living animals	Social Behaviour—examining relationship between personality and bond strength	Behaviour Coding	
Baker, Lea & Melfi (2015)	Sulawesi Crested Macaque
(Macaca nigra)
Barbary Macaque
(Macaca sylvanus)
Squirrel Monkey
(Saimiri sciureus)	Zoo-housed animals	Method Validation—assessing cross-species comparisons of personality	Behaviour Coding + Trait Rating	
Eckardt et al. (2015)	Gorilla
(Gorilla beringei)	Free-living animals	Method Validation—identifying personality structure and behavioural associations	Trait Rating	
Koski & Burkart (2015)	Common Marmoset (Callithrix jacchus)	Laboratory Animals	Social Behaviour—group-level similarities in personality.	Context Tests	
Morton et al. (2015)	Brown Capuchin
(Sapajus apella)	Laboratory animals	Social Behaviour—examining relationship between personality and bond quality	Trait Rating +
Behaviour Coding	
Úbeda & Llorente (2015)	Chimpanzee
(Pan troglodytes)	Sanctuary animals	Method Validation—validating assessment method for use in sanctuary chimpanzees	Trait Rating	
Weiss et al. (2015)	Bonobo (Pan paniscus)	Zoo-housed animals	Method Validation—comparisons of human and bonobo personality factors	Trait Rating	
Altschul, Terrace & Weiss (2016)	Rhesus macaque (Macaca mulatta)	Laboratory animals	Management—associations between personality and intelligence	Trait Rating	
Garai et al. (2016)	Bonobo (Pan paniscus)	Free-living animals	Method Validation—examining factors influencing personality in a free-living primate	Trait Rating + Behaviour Coding	
Molesti & Majolo (2016)	Barbary Macaque
(Macaca sylvanus)	Free-living animals	Social Behaviour—factors affecting cooperation and free partner choice	Context Tests	
Robinson et al. (2016)	Brown Capuchin
(Sapajus apella)	Laboratory animals	Animal Welfare—associations between personality and subjective well-being	Trait Rating	
Šlipogor et al. (2016)	Common Marmoset
(Callithrix jacchus)	Laboratory animals	Method Validation—assessing cross-context consistency in behaviour	Behaviour Coding	
Uher & Visalberghi (2016)	Capuchins (Sapajus spp.)	Laboratory Animals	Method Validation—observations vs. trait rating and biases in assessment methods	Behaviour Coding + Trait Rating	
Wergård et al. (2016)	Cynomolgus Macaque
(Macaca fascilaris)	Laboratory animals	Management—associations between personality and training success	Trait Rating	
Altschul et al. (2017)	Chimpanzee (Pan troglodytes)	Zoo-housed animals	Management—performance and participation in a novel task	Trait Rating	
Blaszczyk (2017)	Vervet Monkey
(Chlorocebus pygerythrus)	Free-living animals	Method Validation—assessment of method validity in predicting personality	Context Tests	
Capitanio et al. (2017)	Rhesus Macaque
(Macaca mulatta)	Laboratory animals	Social Behaviour—longitudinal impact of personality on social pairing success	Context Tests + Trait Rating	
Koski et al. (2017)	Common Marmoset (Callithrix jacchus)	Laboratory animals	Method Validation—Identifying major personality dimensions and possible demographic differences	Trait Rating	
Martin & Suarez (2017)	Bonobo
(Pan paniscus)	Zoo-housed animals	Method Validation—demonstration of novel personality assessment method	Behaviour Coding	
Polgár, Wood & Haskell (2017)	Squirrel Monkey
(Saimiri sciureus)	Zoo-housed animals	Management—individual differences in response to visitors and research participation	Trait Rating	
Račevska & Hill (2017)	Gorilla
(Gorilla gorilla)	Zoo-housed animals	Social Behaviour—examining social behaviour and personality traits	Trait Rating	
Robinson et al. (2017)	Chimpanzee
(Pan troglodytes)	Zoo-housed animals	Animal Welfare—associations between personality and subjective well-being	Trait Rating	
Weiss et al. (2017)	Chimpanzee
(Pan troglodytes)	Free-living animals	Method Validation—comparing temporal reliability of historic and modern ratings	Trait Rating	
Altschul et al. (2018)	Chimpanzee
(Pan troglodytes)	Zoo-housed animals	Animal Health—links between personality and longevity of chimpanzees	Trait Rating	
Gottlieb et al. (2018)	Rhesus Macaque
(Macaca mulatta)	Laboratory animals	Animal Health—relationship between personality, stressors and diarrhea	Context Tests +
Trait Rating	
Inoue-Murayama et al. (2018)	Common Marmoset
(Callithrix jacchus)	Laboratory animals	Animal Welfare—relationship between personality subjective well-being	Trait Rating	
Hopper, Cronin & Ross (2018)	Japanese Macaque
(Macaca fuscata)	Zoo-housed animals	Method Validation—assessing reliability of short-form trait rating assessment	Trait Rating + Context Tests	
Lutz & Brown (2018)	Cynomolgus Macaque
(Macaca fascilaris)	Laboratory animals	Management—personality differences in enrichment (porch) usage	Context Tests	
Masilkova, Weiss & Konečná (2018)	Cotton-top Tamarin
(Saguinus oedipus)	Zoo-housed animals	Method Validation—developing reliable and efficient personality assessment for tamarins	Behaviour Coding	
Robinson et al. (2018)	Rhesus Macaque
(Macaca mulatta)	Laboratory animals	Animal Health—associations between personality, dominance, and health	Trait Rating	
Brandão et al. (2019)	Brown Capuchin (Sapajus apella)	Zoo-housed animals	Method Validation—using behavioural observations in assessments of capuchins	Behaviour Coding	
Ebenau et al. (2019a)	Assamese Macaque
(Macaca assamensis)	Free-living animals	Method Validation—providing baseline for M. assamensis personality research	Behaviour Coding	
Ebenau et al. (2019b)	Assamese Macaque
(Macaca assemensis)	Free-living animals	Social Behaviour—relationship between personality and social bonding in males	Behaviour Coding	
Fernández-Lázaro et al. (2019)	Cotton-top Tamarin
(Sagiunus oedipus)
Emperor Tamarin
(Saguinus imperator)
Golden lion Tamarin
(Leontopithecus rosalia)
Common Marmoset
(Callithrix jacchus)
Geoffrey’s Marmoset
(Callithris geoffroyi)
Pygmy Marmoset
(Cebuella pygmaea)
White-faced Saki
(Pithecia pithecia)
Pygmy Loris
(Nycticebus pygmaea)	Zoo-housed animals	Animal Health—cross-species examination of the relationship between personality, lateralisation and physiological welfare indicators	Context Tests + Trait Rating	
Tomassetti et al. (2019)	Common marmoset (Callithrix jacchus)	Laboratory Animals	Method Validation—verifying presence of personality and comparisons with lateralisation.	Behaviour Coding	
Verspeek et al. (2019)	Bonobo
(Pan paniscus)	Zoo-housed animals	Social Behaviour—relationship between personality and relationship quality	Behaviour Coding	
Costa et al. (2020)	Golden-headed lion tamarin
(Leontopithecus chrysomelas)	Free-living animals	Animal Health—examining links between personality, habitat use and health status	Context Tests	
					
Ruhde et al. (2020)	Rhesus Macaque
(Macaca mulatta)	Laboratory animals	Social Behaviour—outcome predictors for new social groups	Context Tests	

Appendix 2 Trait rating instruments used in reviewed non-human primate personality research, with the most commonly used tools presented first.

Name of Trait Rating Instrument	Proportion of Trait Rating studies (%)	Reference	
Hominoid Personality Questionnaire (HPQ)	41	(Weiss, 2009, 2017)	
Maddingley Questionnaire	13	(Stevenson-Hinde & Zunz, 1978)	
Biobehavioural Assessment (BBA) Temperament Scale	11	(Capitanio et al., 2005)	
Freeman Questionnaire	7	(Freeman et al., 2013)	
Undefined Trait Rating Assessment	7	(Fernández-Lázaro et al., 2019)	
Gorilla Behavioural Index	7	(Kuhar et al., 2006)	
Five Factor Model	2	(Robinson et al., 2017)	
BIAZA Behavioural Profiling Guidelines	2	(Ward & Melfi, 2013)	
Capuchin Personality Inventory	2	(Uher & Visalberghi, 2016)	
Macaque Personality Inventory	2	(Uher, Werner & Gosselt, 2013)	

Appendix 3 Benefits and drawbacks of the principal personality assessment methods utilised in studies of non-human primates, synthesised across all studies.

Method	Benefits	Drawbacks	
Behavioural Coding	- Produces data which is comparable between subjects
- Measurements are objective
- Does not require experience with individual animals	- Time-consuming; can take hours to assess each individual
- May not account for variability due to confounding variables
- Based on observable behaviours in one context	
Context Tests	- High level of control limits the impact of confounders
- Easy to conduct; the test can be standardised across institutions	- Only measures a small selection of traits (e.g. fearfulness, reactivity)
- Not always clear how responses should be measured and interpreted
- Based on a few specific behaviours	
Trait Rating	- Quick, straightforward to complete
- Accounts for variability in behaviour across contexts	- Differences in familiarity with subjects
- Measurements depend on subjective judgement
- Requires validation to confirm surveyed traits correlate with observable behaviours	

Additional Information and Declarations

Competing Interests

Author Contributions

Data Availability

The authors declare that they have no competing interests.

Max Norman conceived and designed the experiments, performed the experiments, analyzed the data, prepared figures and/or tables, authored or reviewed drafts of the paper, and approved the final draft.

Lewis J. Rowden conceived and designed the experiments, authored or reviewed drafts of the paper, and approved the final draft.

Guy Cowlishaw conceived and designed the experiments, authored or reviewed drafts of the paper, and approved the final draft.

The following information was supplied regarding data availability:

All raw data are available in the article and Appendix 1.

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
