# Peer review of "Potential applications of personality assessments to the management of non-human primates: a review of 10 years of study"

_PeerJ, doi:10.7717/peerj.12044_

## Round 0.1 · original submission · Major Revisions

Dear Max,

Thank you very much for your submission, which has now been seen by two reviewers. Both welcome your review and are generally very positive. They do have some suggestions for further improvement, however, which I agree would be useful to consider.

Reviewer 1 suggests shifting focus to the practical implications of personality assessment, rather than a focus on methods; they note that perhaps the skew towards validation is not as pronounced as you indicate, and your paper would be more valuable if practical implications were given greater attention. Personally, I think this is a great suggestion, but it is only a suggestion, and of course, this is your paper, so the overall direction is down to you. I would, however, give it very serious consideration as I think it will broaden the interest of your paper, give it real legs, and increase its value as a contribution to the literature. The body of the review is detailed and thorough, and I won't rehearse all the points again here, but I would like to thank the reviewer for the thoughtfulness of their review.

Reviewer 2, Alexander Weiss (who waived anonymity) suggests that some relevant papers may have been omitted, and comments that a bit more clarity around your inclusion criteria, etc. might be useful if these papers were excluded for a good reason. The other suggestions also largely deal with issues of clarity around definitions and the like, and I agree that a bit more precision and explanation here would pay dividends. Again, I'd like to thank Alexander for his efforts as a reviewer, and also offer our congratulations on the birth of his new baby!

I'm entirely confident you can address these comments, and look forward to seeing a new version soon. Thank you for considering PeerJ as a suitable outlet for your work.

Enjoy the weekend, and all the best,
Lou

Reviewer 1 ·

Basic reporting

The review “Potential applications of personality assessments to the management of non-human primates: A review of 10 years of study” summarizes the literature published between 2010 and 2020 across various fields of research including psychology, animal management, zoology, behavioural ecology, veterinary medicine, and conservation and thus has a broad impact. The review is within the scope of the PeerJ and meets the requirements of the journal. There are however, several issues (or rather suggestions for improvement), which I would like to address.

It has been 11 years since the critical review of Freeman & Gosling (2010) which set the direction and profoundly influenced the field of primate personality. One of the gaps suggested by Freeman & Gosling (2010) was the implementation of the knowledge of individual differences in behaviour of primates into animal management and welfare (3 main areas – enrichment, group composition, reintroduction). The current review reappraises the state of research on the practical implications of primate personality in management practices since the review of Freeman and Gosling (2010). Authors identified and discussed 5 most studied areas of personality applications in primate management (social management, environmental enrichment, training protocols, health and welfare monitoring, and conservation planning). New reappraisal of the field is timely as the last attempt to review this topic was made by Kristine Coleman in 2012 (Coleman K. 2012. Individual differences in temperament and behavioral management practices for nonhuman primates. Appl Anim Behav Sci 137: 106–113) who identified similar areas of interested and suggested recommendations for implementation. Authors might wish to acknowledge the study of Coleman (2012) in their review.

The review is well written and broad. Authors state (L105-107) that the review primarily focuses on zoo management (with applications also in other captive facilities). However, focusing on methods of personality assessment potentially obstructs seeing the progress in the field and the practical implications. The review would benefit from focusing more on the research of practical implications (than on methods) and more in detail on the specific topics of practical implications. Suggesting specific recommendations tailored for zoo (and other captive) management could provide guidelines for the future research and help in management decisions. These issues are addressed and discussed in detail in the following 2 parts.

Experimental design

The current review revealed and the authors state that “the majority of studies (31%)” (L29-30) and that “method validation dominated the literature” (L184-185). This result might seem worrying. However, taken together, studies on animal personality and practical implications (management, welfare, health, conservation) accounted for 45% (69% including social behaviour) (Tab 3). Thus, the opposite is truth, which should be acknowledged.

For their search, the authors used a combination of words “primate” AND (temperament OR personality) AND (welfare OR management OR conservation) and then discarded papers that were not relevant resulting in the list of 58 articles. In this list, there are, however, articles that are not directly connected to welfare, management or conservation. These articles might mention potential implications for welfare, management or conservation, but they don´t evaluate them empirically (8/58 studies describing method of personality assessment: Freeman et al. 2003; Highfill et al. 2010; Úbeda and Llorente 2015; Šlipogor et al. 2016; Martin and Suarez 2017; Weiss et al. 2017, Hopper et al 2018; Masilkova et al. 2018; 3/58 studies describing personality structures: Koski et al. 2011; Morton et al. 2013a, Baker et al. 2015). These studies might be interesting for evaluating the spectrum of species that were studied in captivity (and thus set priorities for future research) but cannot inform about practical implications. Studies on wild primates were also included (12/58 studies covering various topics: Dammhahn and Almeling 2012; Konečná et al. 2012; Seyfarth et al. 2012 Carter et al. 2014; Carter et al. 2012; Pritchard et al. 2014; Seyfarth et a.l 2014; Eckardt et al. 2015; Molesti and Majolo 2016; Blaszczyk 2017; Ebenau et al. 2019a; Ebenau et al. 2019b). Although studies on wild populations can be informative about potential implications (although not tested yet in captivity), they might skew the overview of studied species and general research focus. Moreover, including not relevant studies can cause this relatively high percentage (31%) of methodological studies. Therefore, the authors might wish to revise the selection of the relevant articles (e.g. focusing only on captive species and/or only on topics relevant to practical implications).

It would be also beneficial for the researchers and the field in general to see whether the research on practical implications changed and how since Freeman and Gosling (2010). Specifically, authors could compare the number of studies on primate personality and practical implications (welfare, conservation, management) between 2 periods (2000-2010 vs. 2010-2020) to see whether there is actually a decreasing or increasing trend to study this issue nowadays. Moreover, the authors could evaluate which topics are currently studied and which are understudied and how the field developed (e.g. by comparing the information to Coleman et al. 2012 who introduced very similar areas). New topics are e.g. the social networks, psychological well-being and missing are studies on interactions with caretakes. Thus, authors could give more specific recommendations to set the future direction of research.

For the researchers as well as people involved in the management of zoo populations would be also interesting to see the proportion of studied primate species (and families) in captivity (either in general regarding any personality aspect or specifically in the connection with practical implications) from all primate species in captivity. The authors could easily identify families or groups that are not represented in the research (e.g. gibbons, lemurs, howler monkeys, …) and explain why is that (i.e. difficult to study in captivity due to low numbers, difficult to handle them in zoo settings, or recognize them individually without marking, … ). The authors could specifically mention which groups shall the further research focus on.

Validity of the findings

The review would benefit from focusing more on the practical implications as mentioned above:

One suggestion would be to move the Tab 2 to Supplement (similar table published in Freem and and Gosling 2010 and in Koski SE 2011. How to Measure Animal Personality and Why Does It Matter? Integrating the Psychological and Biological Approaches to Animal Personality. In: Inoue-Murayama M, Kawamura S, Weiss A (eds) From Genes to Animal Behavior. Springer Japan, pp 115–136).

Another suggestion is to include the names of personality dimensions (i.e. Boldness, Sociability, …) and their exact relationship (e.g. +, -, 0) with the response variables in tables 3-7. To safe space, the Setting and Personality measure could be replaced by abbreviations (i.e. Z = zoo, L = lab, …). I suggest to rename “Personality measure” to a more straightforward “Method of personality assessment” or “Personality assessment method”. Knowing that the definition of particular personality dimensions might depend on the method used, the information about which dimensions and how exactly they relate to the animal welfare, conservation, health, … measures could be used to inspire further and more detailed research and could be more easily implemented into the current management practices by captive facilities. Based on this overview, it will be also possible to conclude which personality dimensions are the most important predictors or animal management issues (although generalization should be treated with care).

Finally, I would like to encourage the authors to include a paragraph on possible other (specific) implications relevant to the zoo and other captive management, either based on research done on other non-primate species in captivity or on their professional opinion (based on their practical experience). These implications could inspire further research. Below are some tips:

1. Personality has been connected to reproductive success (e.g. litter size and offspring survival) – this knowledge could be used to increase reproductive success in critically endangered or difficult to breed species. Reproductive success is easy to assess in captivity because of the detailed records - studbooks, breeding reports, pedigree info from ZIMS. See e.g.
Wielebnowski NC (1999) Behavioral differences as predictors of breeding status in captive cheetahs. Zoo Biol 18:335–349
or studies on wild primates (Seyfarth et al., 2012).

2. Personality can affect reproductive partner choice and compatibility.
Martin MS, Shepherdson DJ (2012) Role of Familiarity and Preference in Reproductive Success in Ex Situ Breeding Programs. Conserv Biol 26:649–656. https://doi.org/10.1111/j.1523-1739.2012.01880.x

3. Further ideas might be: personality could help to identify which individuals are suitable to house in mixed-species exhibition, walk-through exposures (e.g. to prevent visitor-animal conflicts), suitable for exposition purposes, suitable for the standard method of capturing (via nets) versus less stressful box capturing, which individuals will habituate faster to new environments, …

4. To encourage research on the link between primate personality and animal welfare, management, etc. authors could suggest methods that could be easily applied in captive facilities and that could promote the research, such as shorter forms of personality questionnaires (such as e.g. in Robinson et al. 2018), installing cameras to for behavioural coding, easy and short form psychological well-being (Weiss et al. 2009), using body scoring form to assess body condition, etc.

The third recommendation in the Conclusion reads as “Future study should address key gaps in the primate personality literature; particularly (i) regarding taxa which are currently underrepresented in studies and (ii) exploring the links between personality and health, welfare, social management, and other practical areas of interest in greater detail.“ The authors can be specific and identify underrepresented taxa and other areas based on their review.

Finally, there are some minor issues within the manuscript:
L150: replace “popular” by “studied”
L154-157: “It is worth noting that“ redundant
L248: please delete “counterintuitively“
L279: “less” should not be in italics
L364-365: please, rephrase so it reads “… high scores on Activity, … “

Additional comments

I wish the authors all the best and I hope that the suggestions above will help to improve this review and spark future research and discussion on the links between personality and welfare.

·

Basic reporting

Review of Potential applications of personality assessments to the
management of non-human primates: A review of 10 years of study
(#56138)

The manuscript was a review of work on personality in nonhuman
primates since 2010. The review focused largely on the utility of
personality assessments of various sorts in the management of captive
primates and, to a lesser extent, primates reintroduced into the
wild. Seeing as 10 years have passed since the previous, more general,
review by Freeman and Gosling, the overall goal of this review is to
be lauded. It's important to review the state of the field, especially
given that work in the area is beginning to move into applications of
these measures.

Before saying anything else, I apologise for taking longer than
expected to complete the review. I recently welcomed another child
into the world, and so things are a bit less settled than they usually
are around here.

I had a few moderate comments and several minor ones that I hope the
authors can address. I'll begin with the former.

1. The review seems to have missed some papers, which I thought should
be there. Perhaps these fell outside the scope of the literature
search, and if so, the search should have been described more clearly,
or they weren't missed by the authors.

One such paper was by Altschul et al. (2017):
https://royalsocietypublishing.org/doi/full/10.1098/rsos.170169

Another was by Weiss et al. (2015)
https://doi.org/10.1177%2F0956797615589933

And there have been empirical papers published by Uher:
http://dx.doi.org/10.1016/j.jrp.2016.02.003
http://dx.doi.org/10.1016/j.jrp.2013.03.006
https://doi.org/10.1016/j.jrp.2013.01.013

I suspect there are other papers, but I will leave the task of finding
them to the authors, unless, of course, these and the other papers
were omitted for reasons that were not made clear (to me).

2. In lines 170-171, the authors cite King and Figueredo as the
originators of the Hominoid Personality Questionnaire (the HPQ). The
HPQ is based on the questionnaire used in their 1997 paper (the
Chimpanzee Personality Questionnaire) in that it is an extension of
this questionnaire. However, the references for the HPQ are:

a). Weiss A. et al. Am J Primatol. 2009. https://doi.org/10.1002/ajp.20649
b). Weiss A. 2017. https://doi.org/10.1007/978-3-319-59300-5_2

Note that the latter reference is a book chapter which outlines the
questionnaire's history and development. I have attached it to my review.

3. In lines 266-272, and perhaps further, the authors refer to
sociability. Here the writing would lead a reader to believe that
sociability is the same as extraversion. This blurs some distinctions
that have been noted, both in the key papers, but also in a review of
the field by Weiss (2017, J Pers. https://doi.org/10.1111/jopy.12310.

Specifically, depending on the species, a single factor (sociability)
loads on social traits or two factors comparable to the Big Five
factors of extraversion and agreeableness load on these traits,
although bonobos (see the 2015 paper cited above) show a different
configuration. I don't think the authors need to get into the nitty
gritty of it all, so they could simply refer to personality factors
associated with social traits, and then just name the different
factors when citing the relevant studies.

4. In line 339 the authors refer to "facets", but it is not clear to
what they are referring to. Are they referring to what are known as
factors, domains, dimensions, for example, extraversion or one of the
other Big Five dimensions in humans, or do they mean lower order
constructs, such as the facets of each of the Big Five, e.g., the
"warmth" facet or extraversion?

Whatever the case may be, the authors need to be consistent in their
terminology here, maybe every so briefly defining any new term.

A helpful reference, by the way, about facets is a 1995 paper by PT
Costa and RR McCrae published in the J Pers Assess, http://doi.org/10.1207/s15327752jpa6401_2

My minor comments are as follows, and I refer to line numbers where I can:

1. In the Abstract, I think I would delete "widely", which is true,
but it is implied by the the rest of the text. In fact, it's a
good idea to omit as many *ly words as possible.

2. I've always known "non-human" to be spelled without a hyphen,
namely as it's not a compound word.

l 42. I would omit "it be"

l 44. "incorporation" -> "incorporating"

l 47. Here and elsewhere the authors use causal language
("predicted"), which is a problem when studies are anything other than
true experiments. I would be more modest and would use terms like
"associated" and "related" instead.

l 55. The authors will want to note that these measures are reliable
and validated. Here I think the authors ought to provide definitions
from the literature on psychometrics and construct validity. A 1955
paper by Cronbach and Meehl (https://doi.org/10.1037/h0040957) and a
1959 paper by Campbell and Fiske (https://doi.org/10.1037/h0046016)
are good for defining validity, but many textbook treatments will be
fine. Reliability has been discussed extensively, too.

l 58. I would omit "existing".

l 59: Is the word "animal" needed here?

l 64: Is the word "validated" necessary here?

L 82: I think the word "numerous" is redundant; it can be omitted.

l 92: Consider omitting this line.

l 97: I would omit the word "then".

l 106: I would change "have generality" to "generalize". Here, too, a
definition would be nice.

l 149: I would omit the word "different".

ll 173-174: I would revise the text so that it reads "...and each has
its benefits and drawbacks" and "which we have outlined".

l 189: If the authors could cite an example cite here, that would be good, too.

ll 214-216: Some examples here would be helpful.

l 217: I think there could be a a better transition sentence at this
point.

l 241: Please cite who proposed this idea.

l 250: I would not refer to "types" when personality traits, factors,
etc. are better thought of as being on a continuum. Thus, refer to
"individuals higher or lower in..."

l 254: Might it be worth considering what the human literature shows?
Is this a promising avenue for future work, in other words?

ll 298-299: I would use the active voice here and throughout.

l 306: I would rewrite this sentence without the text "take a holistic
approach which"

l 326-330: Please shorten or split this sentence.

l 334: I'd rephrase this as "the shy-bold continuum".

l 337: Here I would strive to be consistent in labels that are
used. So if "shyness-boldness" is referred to before, just use that
label again.

l 351: What do the authors mean, in particular, by "in-depth"? That
is, what aspects need to be looked at in more depth?

l 358: I would rephrase to "voluntarily participate".

ll 488-494: I think this section could do with some reference to the
fact that heritable personality variation is the norm and provide some
indication of what mechanisms might support this in wild
populations. That's going to be key with regards to reintroduction
programs and breeding, etc.

l 496: I think the word "some" could probably be deleted.

ll 538-539: For clarity, it's best to phrase this kind of thing as
"shifting" from something "to" something else.

Thank you for asking me to review this interesting paper. I hope the
authors find my comments helpful.

Alexander Weiss [I sign my reviews]

Experimental design

See above.

Validity of the findings

See above.

Additional comments

See above.

---

## Round 0.2 · Minor Revisions

Dear Max

Thanks for your revision, which I have read over myself, and I also asked one of our previous reviewers to take another look. As you will see, they agree that the new version represents a great improvement. Alexander did have a few minor comments and corrections, so if you could just take a look at these and amend your MS accordingly, I'll then be able to move to final acceptance.

All the best,
Lou

·

Basic reporting

N/A

Experimental design

N/A

Validity of the findings

N/A

Additional comments

Review of 56138v2: Potential applications of personality assessments to the management of non-human primates: A review of 10 years of study
This is a revised version of a manuscript that I had previously reviewed. I thought it was a useful review and most of my comments were minor. The authors did a good job in responding to these comments and this version of the manuscript is a clear improvement. I only had some minor comments remaining, most having to do with writing. I list these below in no particular order.
Can omit “disparate and” from the Abstract.
“since 2010”: Note this comes after an earlier review
“increased progression” -> “progression”
“existing animal” -> “animal”
l 46: Perhaps just spell out “for example” throughout but be sure to have it followed by a comma in any event. Same with “i.e.”
l 49: delete “well”.
ll 60-65: Refs to show the validity and reliability studies?
l 89: “carried out” (omit)
ll 104-113: The word “will” comes up many times. Is it needed?
l 144: Is “further” needed?
l 174: Some duplicated text (“, to name a few”)
l 189: Is “observer” needed?
l 205: Insert “single”?
ll 241-242: Omit “an area of research”
ll 305-308: A bit confusing as written. Suggest that it reads “A study by Morton et al. (2015) found that captive capuchin (Sapajus spp.) dyads who were more similar in the Sociability dimension, had higher-quality relationships.”
l 351: “common squirrel monkeys”
l 364: Omit “more”.
l 373: The word “facets” is used here.
l 374: Insert “studying” before “the interaction”
l 375: Delete “approach”?
l 381: The word “type” should be omitted.
l 399: Wouldn’t the Altschul et al. 2017 study fit into this? Same goes for Altschul et al. 2016. (doi: 10.12966/abc.02.04.2016).
l 463: The apostrophe is in the wrong place.
l 498: Can omit “different”.
l 527: In cases like this, shouldn’t an “e.g.” precede the references as this is not an exhaustive list?
ll 526-529: The 50% heritable number comes from a lot of studies on animals, which these authors cite. Also, the statement “is attributable to additive genetic variation and is thus heritable” is somewhat redundant as to be heritable means that variance is due to additive genetic effects. I would just say “is heritable”.
ll 536-537: “a number of” -> “several”.
l 540: The species is cited earlier (line 429) but it’s only now that the scientific name is given. I would move the scientific name to the first reference of this species.
l 551: “approach” not “theory”.
l 567: I believe what was concluded was that one of the assays could not have been measuring boldness.
l 577: Delete “of these”
l 606: Delete “The”.
ll 614-616: I’d be cautious about endorsing short questionnaires. They have numerous problems associated with them, which I won’t go into. Suffice it to say, they may not cover all aspects of some trait and thus their use may provide misleading results (see, e.g., Weiss & Costa 2014, doi: 10.1093/aje/kwt300).
Thank you for asking me to review this interesting manuscript. I hope the authors find my comments helpful.
Alexander Weiss [I sign my reviews]

---

## Round 0.3 · accepted · Accept

Hi Max, Lewis and Guy

Thanks very much for making this last few changes. I'm delighted to accept your paper for publication and hope it receives the attention it deserves. Congratulations on a fine and valuable piece of work!
Cheers
Lou